bioengineering/ecology/synthetic biology

synthetic biology, ecological engineering, climate change, catastrophic shifts, mutualism, synthetic ecology

**Author for correspondence:**
Ricard V. Solé
e-mail: ricard.sole@upf.edu

# Synthetic soil crusts against green-desert transitions: a spatial model

Blai Vidiella[1,2], Josep Sardanyés[3,4] and Ricard V. Solé[1,2,5]

[1]ICREA-Complex Systems Lab, Universitat Pompeu Fabra, 08003 Barcelona, Spain
[2]Institut de Biologia Evolutiva (CSIC-UPF), Psg. Maritim Barceloneta, 37, 08003 Barcelona, Spain
[3]Centre de Recerca Matemàtica, and [4]Barcelona Graduate School of Mathematics (BGSMath), Edifici C, Campus de Bellaterra, 08193 Cerdanyola del Vallès, Bellaterra, Barcelona, Spain
[5]Santa Fe Institute, 1399 Hyde Park Road, Santa Fe NM 87501, USA

 BV, 0000-0002-4819-7047; JS, 0000-0001-7225-5158;
RVS, 0000-0001-6974-1008

Semiarid ecosystems are threatened by global warming due to longer dehydration times and increasing soil degradation. Mounting evidence indicates that, given the current trends, drylands are likely to expand and possibly experience catastrophic shifts from vegetated to desert states. Here, we explore a recent suggestion based on the concept of ecosystem terraformation, where a synthetic organism is used to counterbalance some of the nonlinear effects causing the presence of such tipping points. Using an explicit spatial model incorporating facilitation and considering a simplification of states found in semiarid ecosystems including vegetation, fertile and desert soil, we investigate how engineered microorganisms can shape the fate of these ecosystems. Specifically, two different, but complementary, terraformation strategies are proposed: *Cooperation*-based: *C*-terraformation; and *Dispersion*-based: *D*-terraformation. The first strategy involves the use of soil synthetic microorganisms to introduce cooperative loops (facilitation) with the vegetation. The second one involves the introduction of engineered microorganisms improving their dispersal capacity, thus facilitating the transition from desert to fertile soil. We show that small modifications enhancing cooperative loops can effectively modify the aridity level of the critical transition found at increasing soil degradation rates, also identifying a stronger protection against soil degradation by using the *D*-terraformation strategy. The same results are found in a mean-field model providing insights into the transitions and dynamics tied to these terraformation strategies. The potential consequences and extensions of these models are discussed.

# 1. Introduction

Global warming is changing the dynamics and resilience of ecosystems, damaging many of them and creating the conditions for widespread diversity loss [1–3]. Because of the presence of nonlinearities, many ecosystems could suffer the so-called tipping points [4–6], responsible for community collapses and for transitions towards more degraded states [7,8] (e.g. coral bleaching [9] or lake eutrophication [10]). Among these systems, drylands (which comprise arid, semi-arid and dry-subhumid ecosystems) are a specially fragile subset of major importance: they include more than 40% of terrestrial ecosystems and host a similar percentage of the current human population [11,12]. Increasing aridity is pushing these ecosystems towards serious declines in microbial diversity, land degradation and loss of multifunctionality as desert states are approached [11,13,14]. Dedicated efforts have been addressing several avenues to both understanding how transitions can be anticipated by means of warning signals [15–17] and even prevented [11,18,19]. Catastrophic transitions i.e. the transition that occurs at the tipping point, from vegetated to desert ecosystems can be really fast [19], even in large ecosystems [20]. This type of sudden transitions have been identified in the evolution of the vegetated Sahara to the current desert state, which occurred less than 6000 years ago [21–23].

Drylands are characterized by the presence of organisms that have adapted to low moisture availability, damaging UV radiation, and high temperatures [24–26]. A rich community structure and the maintenance of physical soil coherence are essential to prevent drylands from degradation [27–29]. In this context, some universal types of interactions that occur among species in arid habitats inevitably lead to breakpoints associated with the existence of multiple alternative states, identified in field data [14,30]. A well-known class of these interactions takes place among vascular plants and is known as facilitation [31–33]. Facilitation consists on non-trophic interactions between individuals mediated through changes in the abiotic environment favouring individual growth and reproduction [34–36]. An example of this process is the fertile islands in the savannah ecosystems, where the presence of trees increase the soil organic carbon thus increasing soil moisture and nutrients [33]. This process facilitates the implantation and growth of new vegetation (grasses and trees) [37,38].

A common outcome of spatially extended ecosystems is the emergence of spatial patchiness e.g. peatlands (see [39] and references therein) or the Kalahari desert [38], to cite a few. These spatial patterns have been extensively studied using computational models [39,40], for instance, models only considering competition [41]. Other models including both competition and facilitation processes also typically display self-organized patchiness [38,42–45]. Such patterns are often remarkably organized in space [45]. Transitions between different ecological states, e.g. vegetated towards bare soil, can be sometimes observed in the same landscape under the presence of environmental gradients [31]. In this context, it has been conjectured that spatial correlations can be used as indicators of forthcoming green-desert shifts [15,46,47]. The presence of catastrophic transitions deeply modifies our perception of risks associated with climate change and land degradation. Once a tipping point is reached, unstoppable runaway processes are unleashed [8]. Despite much progress having been made in modelling drylands [47–49] as well as in identifying warning signals [15–17,46], feasible strategies to prevent green-desert transitions are currently scarce.

In a recent paper [42], it has been shown that, by tuning some particular features of models exhibiting catastrophic shifts, one can modify their nature and location in parameter spaces. Based on a theoretical approach, the authors suggested that the amount of stochasticity (both demographic and extrinsic) or population dispersal range can play that role. Indeed, the impact of plant dispersal ranges on the nature of transitions has been recently investigated in a simple metapopulation model with facilitation [43], showing that short-range dispersal might involve continuous transitions. Examples of gradual regime shifts in spatial patterns have been also described theoretically [44]. What type of microscopic mechanisms could actually avoid green-desert transitions? In this paper, we are extending previous models of dryland dynamics [19,47] to predict the impact of bioengineering strategies on these fragile ecosystems. In this context, several bioremediation approaches have been developed in the last two decades to enhance and stabilize soil biocrusts (for an overview, see [26] and references therein). These include a diverse range of approaches addressed to improve moisture and soil texture, from enrichment of key nutrients [50] or enrichment by cyanobacteria [51–53], to large-scale straw chequerboard barriers used to stabilize sand dunes [54].

More recently, ecosystem terraformation has been suggested as a novel approach against green-desert shifts [55–59]. In a nutshell, some existing microorganisms in a degraded ecosystem, such as cyanobacteria of the soil crust, could be minimally modified by means of synthetic biology techniques

to help improve soil moisture and create a cooperative feedback between vegetation or moss cover (see [60]) and the soil microbiome [61]. This engineering scenario aims at building synthetic soil composed of synthetic strains mixed with natural microorganisms embedded, for example, in the soil crust. By doing so, potential tipping points could be made much more difficult to be achieved [58]. What might be the impact of these strategies on the large-scale, long-term dynamics of semiarid ecosystems? How can new or enhanced ecological interactions (cooperation and microbes dispersal capacities) modify the presence of tipping points? As shown below by means of both mathematical and computational models, the engineering of microorganisms present in arid and semiarid soils may easily expand the potential of ecosystems' persistence.

# 2. Modelling terraformation of semiarid ecosystems

In order to predict the impact of synthetic bioengineering strategies in arid and semiarid ecosystems, a spatial model given by a stochastic cellular automaton (CA) is employed. Let us first consider the microscopic rules associated with the dynamics, as described by a set of probabilistic transitions among the three defined ecological states. After this general description of the transitions, we will investigate a mean-field model without stochasticity and the stochastic CA. The mean-field model will allow us to identify the nature of the tipping points and the expected equilibria (which may be useful for low stochasticity levels i.e. large system's size).

The basic transition diagram is displayed in figure 1$c$, which includes the three potential states characterizing a given patch, namely

$$\Sigma = \{D, S, V\}.$$

These states, as mentioned, are defined by desert ($D$) patches, by fertile soil ($S$) composed by patches occupied by engineered microorganism(s) with their natural community (e.g. soil crust), and by vegetation ($V$), respectively. This is, of course, an oversimplification that ignores most of the complexity and diversity involved, but allows for an analysis of the dynamics considering key ecological interactions such as transitions between states and both competition and facilitation processes. The model described here is derived from the one proposed by Kéfi *et al.* [47], now considering that unoccupied sites (non-degraded and non-vegetated) contain soil crust plus (engineered) microorganisms. As discussed in a previous work [58], we aim at describing how an appropriate synthetic modification of microorganisms could help to maintain the stability and resilience of semiarid ecosystems.

The first rule, which considers the transition from degraded to fertile soil containing synthetic strains, will take place with a probability

$$P(D \rightarrow S) = r + f\rho_v + \Gamma(\rho_s).$$

Here, $r$ is the probability of spontaneous recovery of fertile soil due to external factors such as increased humidity, accumulation of organic matter, etc. Since we are interested in identifying the impact of the engineered strains in the recovery of the desert soil to the fertile one we will set $r = 0$. This will allow us to provide clearer results concerning soil recovery derived from terraformation. Constant $f$ denotes the facilitation due to the influence of the surrounding vegetation. The variable $\rho_v$ is the local density of vegetation (number of vegetated neighbours over the total number of neighbours). Finally, $\Gamma(\rho_s)$ denotes the spreading capacity of the microorganisms associated with their synthetic engineered properties (see below).

The second transition considers the reverse situation, namely a fertile soil-to-desert transition

$$P(S \rightarrow D) = \varepsilon\, Y(\rho_v),$$

where $\varepsilon$ is the probability of loss of fertile soil due to increased aridity or to other soil degradation processes. In this paper, we consider (see below) a modification of the degradation rate mediated by a vegetation-dependent function $Y(\rho_v)$. The third transition rule involves the colonization of available $S$ patches by vegetation

$$P(S \rightarrow V) = (\delta\Delta_V + (1 - \delta)\rho_v)(b - c\Delta_V). \tag{2.1}$$

This term has the same form as the colonization used in [47]. The probability $\delta$ balances the influence of the local and global vegetation to produce the germination of new plants in a given site. Seeds are produced and germinate with probability $b$. However, they can also be degraded during the

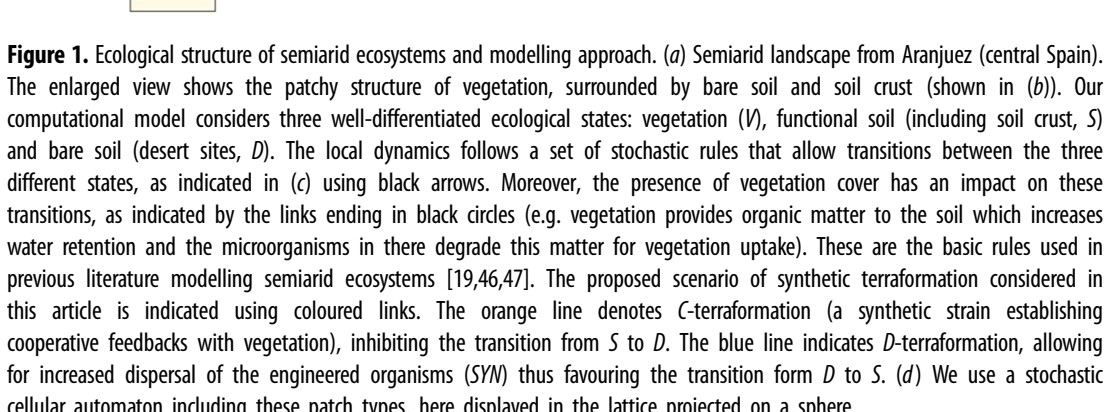

**Figure 1.** Ecological structure of semiarid ecosystems and modelling approach. (*a*) Semiarid landscape from Aranjuez (central Spain). The enlarged view shows the patchy structure of vegetation, surrounded by bare soil and soil crust (shown in (*b*)). Our computational model considers three well-differentiated ecological states: vegetation (*V*), functional soil (including soil crust, *S*) and bare soil (desert sites, *D*). The local dynamics follows a set of stochastic rules that allow transitions between the three different states, as indicated in (*c*) using black arrows. Moreover, the presence of vegetation cover has an impact on these transitions, as indicated by the links ending in black circles (e.g. vegetation provides organic matter to the soil which increases water retention and the microorganisms in there degrade this matter for vegetation uptake). These are the basic rules used in previous literature modelling semiarid ecosystems [19,46,47]. The proposed scenario of synthetic terraformation considered in this article is indicated using coloured links. The orange line denotes *C*-terraformation (a synthetic strain establishing cooperative feedbacks with vegetation), inhibiting the transition from *S* to *D*. The blue line indicates *D*-terraformation, allowing for increased dispersal of the engineered organisms (*SYN*) thus favouring the transition form *D* to *S*. (*d*) We use a stochastic cellular automaton including these patch types, here displayed in the lattice projected on a sphere.

dispersion (with probability *c*). Finally, an exponential decay of vegetated patches to fertile soil occurs following the transition rule

$$P(V \rightarrow S) = m,$$

where *m* is the death probability of plants, which can occur due to pathogens, grazing, or to the recollection by livestock activity.

Here two different, but complementary, terraformation strategies are considered and weighted by means of constants $\alpha$ and $\beta$. Their role within the transition diagram is displayed in figure 1*c*. They involve:

1. *C*-terraformation: Engineering cooperative loops between vegetation and soil organisms. Here, a decreasing function of the vegetation cover, namely

$$Y(\rho_v) = \frac{1}{1 + \alpha \rho_v},$$

   introduces a reduction in the impact of desertification due to increased soil quality (by e.g. nutrient deposition) favoured by the synthetic microbial population. The efficiency of this term is weighted by the constant $\alpha$: large values of $\alpha$ imply a lower soil degradation rate due to the action of the synthetic microbes which are cooperatively coupled with vegetation at a local scale.

2. *D*-terraformation: Engineering the capacity of microbial spreading (e.g. faster replicative rates of the microbes and/or increased formation of endospores able to successfully colonize local surroundings).

The dispersal is here considered proportional to the term

$$\Gamma(\rho_s) = \beta \rho_s.$$

Here, $\rho_s$ is the local density of fertile soil. In other words, with probability $\beta$, engineered strains improve their dispersal capabilities, thus changing the properties of desert patches by retaining humidity, depositing organic matter, etc.

Setting $\alpha$ and $\beta$ to zero, the CA model recovers the original model introduced by Kéfi *et al.* [47].

# 3. Models and results

In the next sections, we will explore the dynamics tied to the rules described above using both a well-mixed (mean-field) and a discrete spatial setting using a stochastic Cellular Automaton (CA). The first approach, developed in §3.1, employs differential equations to predict the presence of tipping points and changes in their location in the parameter space due to the effects of the addition of a synthetic strain. The second approach, studied in §3.2, explicitly deals with a spatially extended population where local interactions take place on a two-dimensional lattice and the set of rules are applied in a probabilistic manner, thus considering stochastic effects. Finally, we provide insights into the nature of the tipping points found in the spatial model, focusing on the early warning signals for the system without and with the synthetic engineered microorganisms.

## 3.1. Mean-field model

The differential equations model, which includes the interactions and processes tied to the bioengineered synthetic strains, is given by

$$\frac{dV}{dt} = V((b - cV)S - m), \tag{3.1}$$

$$\frac{dS}{dt} = (fV + \beta S)D - \varepsilon\, S\, Y - V((b - cV)S - m) \tag{3.2}$$

and

$$\frac{dD}{dt} = \varepsilon\, Y S - (r + fV + \Gamma)D. \tag{3.3}$$

Equation (3.1) contains a logistic term that includes variable $S$ as a multiplicative term, indicating that plants need viable soil to persist, and an exponential decay proportional to $m$. Equation (3.2) includes the positive effects triggered by vegetation and the existing soil cover, as well as negative terms that are in fact symmetric to those present in the previous equation for $V$. Here recall that $Y = 1/(1 + \alpha V)$. Equation (3.3) provides the dynamics of desert patches resulting from desertification, recruitment to viable soil patch, and an exponential decay from $S$ to $D$ introduced with $\Gamma = \beta S$.

Since the three classes of patches cover the entire lattice in the spatial model, it is possible to normalize and consider the states as population fractions in the mean field approach, i.e. $V(t) + S(t) + D(t) = 1$. This actually assumes a constant amount of available sites that can transition between them, also allowing to reduce the three-variables system to a two-variables one using the linear relation

$$D(t) = 1 - V(t) - S(t), \tag{3.4}$$

which needs to satisfy $dV/dt + dS/dt + dD/dt = 0$ (recall that $r = 0$) since the sum of all states remains constant through time. The reduced system is thus given by

$$\frac{dV}{dt} = V((b - cV)S - m) \tag{3.5}$$

and

$$\frac{dS}{dt} = (fV + \beta S)(1 - S - V) - \varepsilon\, S\left(\frac{1}{1 + \alpha V}\right) - V((b - cV)S - m). \tag{3.6}$$

Note that now the fraction $D$ is automatically obtained from equation (3.4) once the fractions of states $V$ and $S$ are determined. We must note that equations (3.5) and (3.6) have been recently studied in [19] for the non-engineered system (that is, taking $\alpha = \beta = 0$).

We are specially interested in those scenarios involving a full dominance of the desert state, focusing in the transitions between states. We have identified four equilibria for equations (3.5) and (3.6), two of

them implying the extinction of vegetation and another one allowing for the stable coexistence of the three ecological states. The first equilibrium is at the origin, i.e. $P_1^* = (V = 0, S = 0)$, were the system becomes a desert ($D = 1$). The second equilibrium, keeps the fertile soil and the desert patches without vegetation, and is given by $P_2^* = (V = 0, S = 1 - \varepsilon/\beta)$. This state is interesting from the point of view of the terraformation strategies, since its existence depends on the chosen engineering strategy. If $\beta = 0$, when no engineered organisms are found in the soil crust, this equilibrium is not biologically meaningful ($S(t \to \infty) = -\infty$).

Two more equilibrium points, labelled $P_3^*$ and $P_4^*$, have been identified numerically (see electronic supplementary material, figures S1–S4 and S6–S8). The equilibrium $P_4^*$ (numerical results suggest it is always stable within the simplex; see below) involves $V(t \to \infty) > 0$, $S(t \to \infty) > 0$ and $V(t \to \infty) + S(t \to \infty) < 1$, allowing the coexistence of the three ecological states when it is an interior equilibrium. $P_3^*$ is a saddle point. See below and electronic supplementary material, figures S1–S4 for further details on the dynamics of the equilibria on the simplex $(V, S)$.

The conditions defining the local stability of each equilibrium can be obtained by means of the eigenvalues of the Jacobian matrix $\mathcal{J}$, given, for equations (3.5) and (3.6), by

$$\mathcal{J} = \begin{pmatrix} -m + (b - 2cV)S & V(b - cV) \\ m - f(-1 + 2V + S) + S(-b + 2cV + \varepsilon\alpha(1 + V\alpha)^{-2} - \beta) & -V(b + f - cV) - \varepsilon(1 + V\alpha)^{-1} + \beta - (V + 2S)\beta \end{pmatrix}.$$

The eigenvalues of matrix $\mathcal{J}$ evaluated at an equilibrium provide its local (and linear) stability properties. The stability for equilibria $P_{1,2}^*$ can be easily computed, while the stability of equilibria $P_{3,4}^*$ will be characterized numerically. In the case of the desert equilibrium $P_1^* = (V = 0, S = 0)$, the eigenvalues are $\lambda_1 = -m$ and $\lambda_2 = \beta - \varepsilon$. Since all the parameters are positive, this equilibrium will be stable when $\varepsilon > \beta$. For $P_2^* = (0, 1 - \varepsilon/\beta)$, its eigenvalues are $\lambda_1 = \varepsilon - \beta$ and $\lambda_2 = -m + b(1 - \varepsilon/\beta)$; see electronic supplementary material, figure S5 for a stability diagram for $P_2^*$ in the parameter space $(\varepsilon, \beta)$. Note that $P_1^*$ and $P_2^*$ suffer a transcritical bifurcation at $\beta = \varepsilon$, since the two conditions for this bifurcation are fulfilled: the two fixed points collide at the bifurcation value (at $\varepsilon = \beta$, $P_1^* = P_2^* = (0, 0)$), and they interchange the stability.

Figure 2 displays the critical degradation rate of the fertile soil, $\varepsilon_c$, computed numerically in the parameter space $(\alpha, \beta)$, separating two domains with and without vegetation at equilibrium. The value of $\varepsilon_c$ moves to higher values as either $\alpha$ or $\beta$ are increased. The yellow region in the surface of $\varepsilon_c$ in figure 2a denotes those values of $\alpha$ and $\beta$ where $\varepsilon_c \geq 1$ (although they are all set to $\varepsilon_c = 1$ for further comparison with the probabilistic results of §3.2). This region denotes that no critical degradation is achieved under the chosen parameter values and under the restriction $0 \leq \varepsilon \leq 1$. The four panels in figure 2b (corresponding to the values of $\alpha$ and $\beta$ indicated in figure 2a with the same letters) provide a more detailed picture of the changes in the fraction of the states due to both C- and D-terraformation strategies. Panel (b.a) shows how the states change without terraformation ($\alpha = \beta = 0$) at increasing $\varepsilon$. Note that for $\varepsilon > 0.2$ the desert state becomes dominant. This critical degradation rate is displaced to larger values for the terraformed system, meaning that the ecosystem becomes much more resistant to soil degradation. More specifically, $\varepsilon_c \approx 0.3$ when applying the C-terraformation strategy (panel (b.b) in figure 2b). This effect is further amplified for the D-terraformation, resulting in $\varepsilon_c \approx 0.7$ (see panel (b.c) in figure 2b). The combination of both terraformation strategies (see figure 2b panel (b.d)) further increases the size of non-desert phases, here with $\varepsilon_c \approx 0.8$. In both cases (specially for D-terraformation), the domain of fertile soil is increased due to the presence of the synthetic strains (electronic supplementary material, figures S6–S8 show how the values of $\varepsilon_c$ are displaced for different intensities of the proposed terraformation strategies).

The dynamics tied to increases of soil degradation rate are summarized in figure 2c (see also electronic supplementary material, figures S1–S4), where different phase portraits are displayed for the values of $\varepsilon$ indicated in figure 2b panel (b.c) (from (i) to (iv)). The transition from the coexistence of vegetation and fertile soil to no vegetation is catastrophic, and is due to a saddle-node bifurcation between fixed points $P_3^*$ and $P_4^*$ (transition between phase portraits (ii) and (iii), see also electronic supplementary material, figures S1–S4). Once this bifurcation takes place, the only stable point is $P_2^*$ (with fertile soil and desert areas). Then, further increase of $\varepsilon$ involves the transcritical bifurcation previously mentioned, after which the origin becomes globally stable and the desert becomes the only possible state (transitions between phase portraits (iii) and (iv) of figure 2c), see also electronic supplementary material, figure S7 where the transcritical bifurcation is shown.

Finally, electronic supplementary material, figure S6 provides one-dimensional bifurcation diagrams tied to the increase of $\varepsilon$ for the non-terraformed system and the three possible terraformation strategies

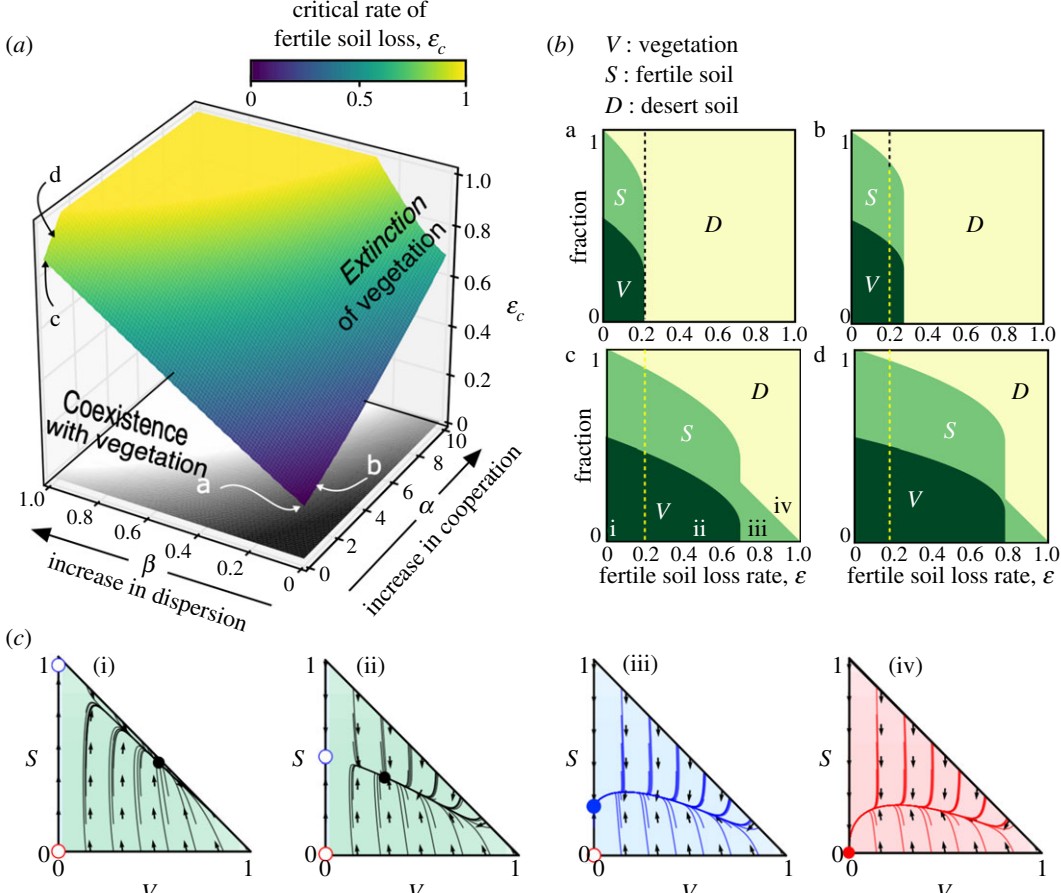

**Figure 2.** Dynamics of the mean-field model for the non-terraformed ecosystem and the terraformed one. (*a*) Critical degradation rate of fertile soil ($\varepsilon_c$) computed in the parameter space of engineering strategies ($\alpha, \beta$) from equations (3.5) and (3.6). The inner surface separates the values of $\varepsilon$ allowing for the presence of vegetation. The yellow area indicates those pairs of $\alpha$ and $\beta$ giving $\varepsilon_c \geq 1$ (those values of $\varepsilon_c > 1$ are set to $\varepsilon_c = 1$). The region with no extinction of vegetation is also displayed projected on the ground of the plot (white zone). (*b*) Fraction of the states at equilibrium increasing $\varepsilon$ and using a full vegetated system as initial conditions $V(0) = 1$, $S(0) = D(0) = 0$. (b.a) Results for a non-engineered ecosystem ($\alpha = \beta = 0$), and (b.b) for the engineered ecosystem incorporating cooperation loops ($\alpha = 1$ and $\beta = 0$). Finally, panels (b.c) and (b.d) display, respectively, the results engineering the resilience of the soil crust ($\alpha = 0$ and $\beta = 1$) and both strategies ($\alpha = \beta = 1$). The dashed vertical lines indicate the critical value of fertile soil loss obtained for the non-engineered system $\varepsilon_c = 0.218$ [19]. (*c*) Phase portraits for the case $\alpha = 0$ and $\beta = 1$, with: (i) $\varepsilon = 0.0$; (ii) $\varepsilon = 0.2$; (iii) $\varepsilon = 0.6$; and (iv) $\varepsilon = 0.9$. The circles indicate the fixed points (stable: solid; unstable: open). We note that in the phase portraits (i) and (ii) there exists an interior saddle (see electronic supplementary material, figures S1–S4 for the identification of the nullclines and dynamics). Green, blue and red regions of the phase portraits denote the equilibrium states of *V-S* coexistence, *S-D* coexistence, and full desert, respectively. The other parameters are $r = 0$, $f = 0.9$, $\delta = 0.1$, $b = 0.6$, $c = 0.3$ and $m = 0.15$.

(*C*-, *D*- and (*C, D*)-terraformation). Here, we want to emphasize the presence of so-called delayed transitions (also called ghosts), which arise just after a saddle-node bifurcation takes place [62]. This is a dynamical phenomenon that involves extremely long transients once the bifurcations has occurred, and the time trajectories experience a long bottleneck before rapidly achieving, in equations (3.5) and (3.6), another attractor (the full desert state in electronic supplementary material, figure S6(a) and (b)). These long transients [63] are typically found in systems with strong feedbacks, such as cooperation, catalytic processes [64,65] or metapopulations with facilitation [43]. Also, this dynamical delay tied to saddle-node bifurcations has been recently described in both deterministic and stochastic well-mixed approaches for the non-terraformed system explored in this article (see [19] for more details).

## 3.2. Spatial stochastic model

The mean-field model studied in the previous section provides a first approximation to understand the qualitative dynamics arising from the nonlinear interactions of the studied system and the resulting tipping points, especially when including the proposed terraformation strategies. However, in order to

test the robustness of these results in a more ecologically realistic setting, one has to take into account both local spatial correlations and stochastic effects [66]. To do so, we build a spatially explicit simulation model given by a stochastic cellular automation (CA). The CA incorporates the previously studied states ($V, S, D$) and transitions among them, which are now probabilistic. Spatial degrees of freedom are introduced by using a $\mathcal{L} \times \mathcal{L}$ square lattice with periodic boundary conditions. At each time step, the following four transition probabilities defining the rate of change for each site $k$ of the lattice (figure 1) are applied in an asynchronous manner:

$$P(D_k \rightarrow S_k) = r + f \frac{1}{\mathcal{M}_{\mathcal{N}}} \sum_{\mu \epsilon \mathcal{M}_k} V_\mu + \beta \frac{1}{\mathcal{M}_{\mathcal{N}}} \sum_{\mu \epsilon \mathcal{M}_k} S_\mu,$$

$$P(S_k \rightarrow D_k) = \frac{d}{1 + (\alpha/\mathcal{M}_{\mathcal{N}}) \sum_{\mu \epsilon \mathcal{M}_k} V_\mu},$$

$$P(S_k \rightarrow V_k) = \left( \frac{\delta}{\mathcal{L}^2} \sum_{\mu \epsilon \Omega} V_\mu + \frac{(1-\delta)}{\mathcal{M}_{\mathcal{N}}} \sum_{\mu \epsilon \mathcal{M}_k} V_\mu \right) \times \left( \frac{b - c}{\mathcal{L}^2 \sum_{\mu \epsilon \Omega} V_\mu} \right),$$

$$P(V_k \rightarrow S_k) = m.$$

Here, $\mathcal{M}_k$ is the neighbourhood of site $k$ and $\mathcal{M}_{\mathcal{N}} = 8$ the number of neighbour (using a Moore neighbourhood). The nature of local (spatial) interactions is known to largely influence ecological dynamics [40,43,67]. This is particularly important in drylands, where carbon and water limitation deeply constrains the outcome of nonlinear exchanges [14,59], leading to spatial patterning [48,68]. Also, facilitation processes (involving strong nonlinearities) are known to introduce important changes in spatial systems, as compared with well-mixed ones. In this sense, recent research has found a shift from catastrophic tipping points to continuous ones, due to local spatial processes [42,43].

Similar analyses to those presented in figure 2 for the mean-field model are displayed in figure 3 for the CA simulations. For the sake of comparison, we have kept the same parameter values of the mean-field model, implemented as probabilities in the CA, also using as initial conditions a full vegetated lattice. The critical surface separating the parameter scenarios allowing the persistence of vegetation is displayed in figure 3a. Below the surface, $V$ persists, while above the surface it is only possible to find states $S$ and $D$. Here, similarly to the mean-field model, the yellow region indicates those values of $\alpha$ and $\beta$ for which no critical degradation rate of the fertile soil is found (i.e. $\varepsilon_c = 1$). Note that the surface for the spatial system differs from the one obtained with the mean-field model. This effect, taking into account that the lattice has $4 \times 10^4$ sites (large system size), is probably introduced by space more than by stochasticity. This change of the surface is especially visible for parameter $\alpha$. This may be explained because soil degradation rate ($\varepsilon$) makes the vegetation decrease globally, but due to the local interactions of facilitation [33], plants remain in the ecosystem for larger rates of soil degradation.

The fraction of the three states computed at increasing $\varepsilon$ is displayed in figure 3b. All diagrams (b.a–b.d) display the same phenomenon: the introduction of spatial correlations shifts the extinction of the vegetation to larger values of $\varepsilon$. Consistently with mean-field results, both the $C$- and $D$-terraformation displace the value of $\varepsilon_c$ to larger values, the ($C, D$)-terraformation strategy being much more efficient in doing so under the selected parameters. The spheres displayed in figure 3c show the spatial patterns at equilibrium for the regions labelled in panel (c) of figure 3b, with: (i–ii) coexistence of the three states; (iii) only fertile soil and desert patches; and (iv) the full desert scenario.

In order to identify the nature of the transitions tied to the increase of $\varepsilon$, we have computed the stationary values of the fraction of areas with vegetation and fertile soil. We recall that both mean field and well-mixed stochastic simulations for the system without terraformation revealed a catastrophic transition and delayed transitions due to a ghost [19]. Also, the mean-field model studied in §3.1 has revealed the presence of saddle-node bifurcations responsible for vegetation extinctions. Figure 4 shows these results together with the spatial patterns on the square lattice and time series. Specifically, figure 4a shows results increasing $\varepsilon$ for the system without terraformation. For this case $\varepsilon_c \approx 0.24$. The same results are displayed by setting $\alpha = \beta = 1$, for which the critical degradation moves to $\varepsilon_c \approx 0.85$. For both cases, the discontinuity of the transition is not so evident as in the mean-field model, even looking like a continuous one.

To identify the transition governing vegetation extinction, we have used an indirect method analysing the properties of the dynamics close to the transition value. The results displayed in electronic supplementary material, figure S9 indicate the presence of bottlenecking phenomena tied to the

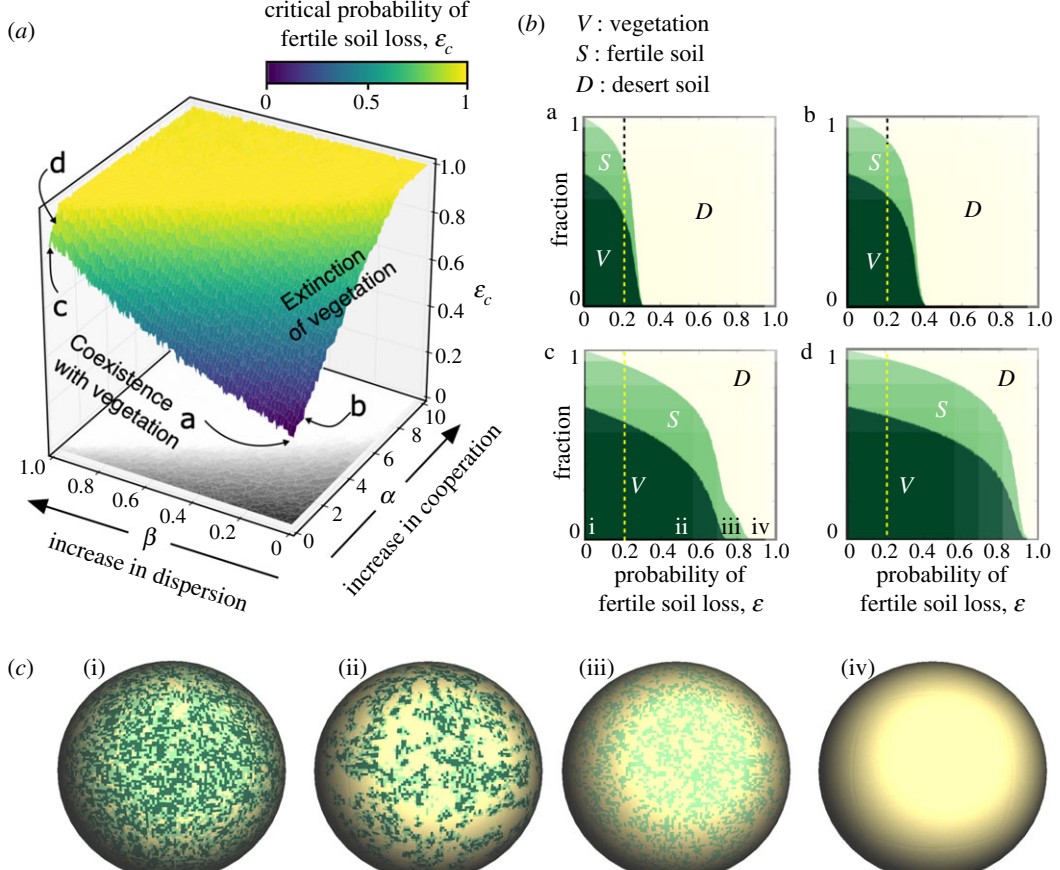

**Figure 3.** Dynamics of the spatial model (using a 200 × 200 sites lattice) with and without terraformation strategies (see figure 2 for comparison). (*a*) Vegetation extinction surface depending on the engineering strategies $\alpha$ and $\beta$. The yellow region indicates that vegetation extinction cannot be achieved because $\varepsilon = 1$ (this region is also projected on the ground of the plot (white zone)). (*b*) Bifurcation diagram using the probability of fertile soil loss $\varepsilon$ for different engineering strategies ($\alpha$ $\beta$). Here, the dashed vertical lines also indicate the critical value of fertile soil loss obtained with the mean-field model for the non-engineered system, $\varepsilon_c = 0.218\cdots$. (*b*.a) Non-engineered ecosystem ($\alpha = \beta = 0$), (*b*.b) engineering of cooperative loops between synthetic microorganisms and the vegetation ($\alpha = 1$ and $\beta = 0$), (*b*.c) engineering resilience of the soil crust ($\alpha = 0$ and $\beta = 1$); and (*b*.d) engineering both terraformation strategies ($\alpha = \beta = 1$). (*c*) Spatial patterns for the case $\alpha = 0$ and $\beta = 1$ (panel (*b*.c) above) with: (i) $\varepsilon = 0.00$; (ii) $\varepsilon = 0.55$; (iii) $\varepsilon = 0.75$; and (iv) $\varepsilon = 0.90$. The probability values have been fixed as the parameters of figure 2, also with $V(0) = 1$.

extinction of the vegetation, meaning that the transition is governed by a saddle-node bifurcation. We must notice that we have used a lattice of side size $\mathcal{L} = 50$ for these analyses due to the huge computational cost of computing extinction transients in large spatial systems. Electronic supplementary material, figure S9(a) shows the transition for the non-engineered system. Specifically, panel (a.1) displays five realizations obtained with $\varepsilon = 0.26$, and the bottleneck region can be clearly seen (see electronic supplementary material, figure S6 for comparison with the mean field bottlenecks). The delaying effect of the ghost can also be seen in panels (a.2), (a.3) and (a.4), which indicate the values at which vegetation spends longer iterating before achieving the full desert state. The same results are shown for the system with $(C, D)$-terraformation. Here, the extinction dynamics of the vegetation (investigated setting $\varepsilon = 0.86$) is also given by a clear bottleneck. Previous research in catalytic hypercycles also revealed extinctions due to saddle-node bifurcations and the corresponding bottlenecking phenomena in a stochastic CA model [65].

As discussed above, the approach to tipping points is often associated with a qualitative change in the nature of the fluctuations exhibited by the system. How will our model behave when dealing with an extra phase associated with vegetation-free, synthetic soil crust? A divergence in a stochastic dynamical system can be detected by using standard deviation measures. We have used several measures that can detect the approach to criticality in both the non-manipulated and the terraformed scenarios (spatial variance and distribution of patch sizes). The spatial variance computed as standard

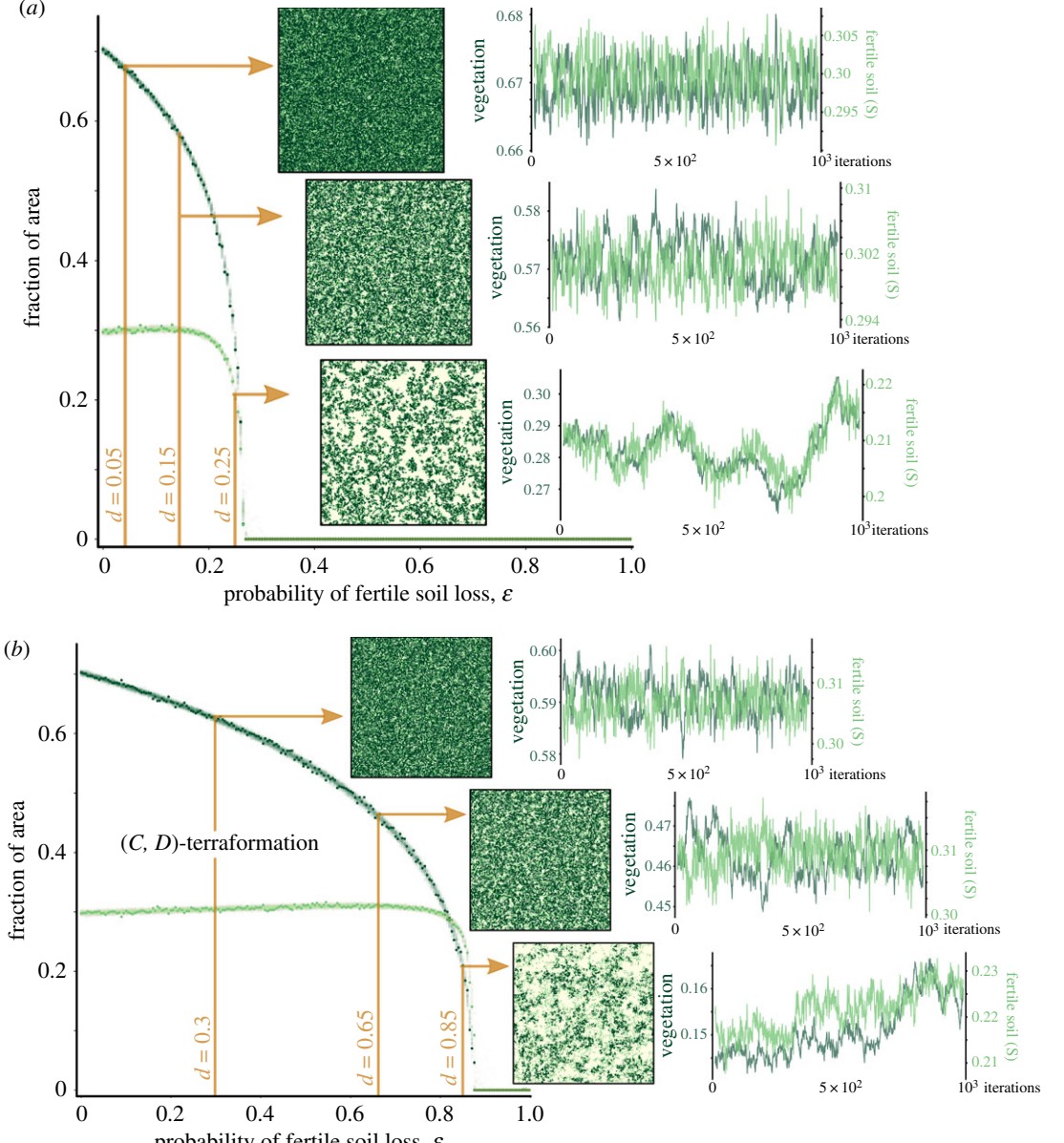

**Figure 4.** Equilibrium population values for the vegetation ($V$) and the fertile soil ($S$) at increasing $\varepsilon$ obtained from the cellular automaton (CA) using a lattice with $200 \times 200$ sites. For each value of $\varepsilon$, the fraction of areas are plotted taking the last 200 values from time series with $10^6$ generations repeated over 10 realizations (to illustrate the variance of the fluctuations). The darkest dots correspond to the last population value of each realization. In ($a$), we display the results for the non-engineered system (with $\alpha = \beta = 0$). Three examples of the spatial patterns at the end of a single realization are displayed for three values of $\varepsilon$, together with time series (using the same colours as in the spatial patterns) obtained at the end of the simulations. ($b$) The same information as in ($a$) for the engineered system ($\alpha = 1$ and $\beta = 1$). Note the huge displacement of the critical soil degradation probabilities between panels ($a$) and ($b$).

deviation $\sigma_\psi$, with $\psi$ referring to either $V$ or $S$, is determined from

$$\langle \sigma_\psi \rangle = \left( \frac{1}{\mathcal{L}^2} \sum_k^{\mathcal{L}^2} (\psi(k, t) - \langle \psi(t) \rangle)^2 \right)^{1/2}, \tag{3.7}$$

where we have used the mean over the entire lattice, given by

$$\langle \psi(t) \rangle = \frac{1}{\mathcal{L}^2} \sum_k^{\mathcal{L}^2} \psi(k, t). \tag{3.8}$$

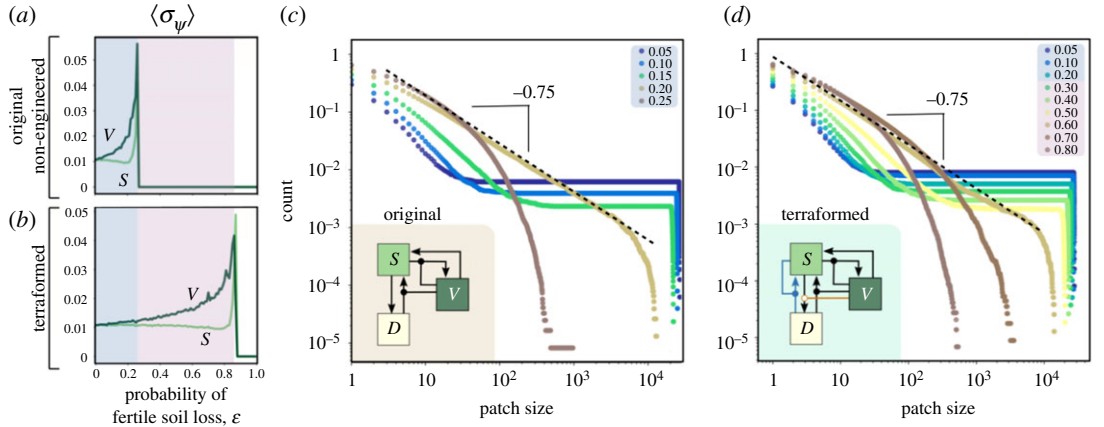

**Figure 5.** Warning signals for the non-engineered ($\alpha = \beta = 0$) and terraformed ($\alpha = 1$ and $\beta = 1$) ecosystems taking vegetation and fertile soil as indicators at increasing $\varepsilon$. Panels (*a–b*) display the standard deviation $\langle \sigma_{\psi=V,S} \rangle$ of the population fluctuations for the original and the terraformation scenarios, corresponding to the simulations shown in electronic supplementary material, figure S9. Values of $\langle \sigma_\psi \rangle$ have been obtained from the last $10^3$ values from time series with $10^6$ iterations. (*c–d*) Cumulative distributions of patch sizes for different values of $\varepsilon$ (indicated with different colours inside the plots). The patch size distributions have been computed using 100 replicas from a lattice with $200 \times 200$ sites.

The fluctuations diverge in a characteristic fashion, as figure 5*a,b* shows: whereas $\langle \sigma_V \rangle$ slowly grows and accelerates close to criticality, the same measure for $\langle \sigma_S \rangle$ displays a slight decline only growing very quickly close to $\varepsilon_c$. Despite these trends, both variables exhibit high fluctuations close to the transition to the desert state.

An additional confirmation of the criticality associated with the transition to the desert state is obtained by determining the distribution of vegetation cluster sizes. In a nutshell, criticality is known to be linked to power law distributions of connected clusters. One particular instance of these scaling behaviours was found in [47]. Clusters of size $S$ are composed by a set of $S$ connected sites (i.e. sharing the same state and all elements in contact with another as nearest neighbours).

The number of clusters of a given size and their distributions have been obtained using a burning algorithm [69]. If $P(S)$ indicates the probability of finding a cluster of size $S$, most measured patch size distributions found in arid and semiarid habitats follow a general form described as a truncated power law, namely $P(S) \sim S^{-\tau} \exp(-S/S_c)$. This defines a power law decay with a scaling exponent $\tau$ that characterizes the statistical organization of systems close to percolation [69] along with an upper bound introduced by a cut-off $S_c$ in the exponential decay term. The cut-off defines the crossover from a regime of critical clusters (power-law distributed) to that of non-critical clusters. As we get closer to percolation points, the system also displays divergent correlation lengths and $S_c \rightarrow \infty$ thus rapidly increasing the cut-off value and moving closer to pure power laws. Since the lattice is finite, we will always observe a rapid decay at the end of the distribution. These distributions can be systematically estimated from vegetation data [38,70,71]. Close to critical transitions we should expect a fat-tailed decay in $P(S)$ following the power law form $P(S) \sim S^{-\tau}$ (when no characteristic scale is present, i.e. for $S_c \rightarrow \infty$). Instead, far below the transition point, the exponential term dominates and a characteristic scale is observed. In order to improve the smoothness of the distribution, the so-called cumulative distribution $P_>(S)$ will be used, and is given by

$$P_>(S) = \int_{S_0}^{S_m} P(S)\, dS, \tag{3.9}$$

where $S_0$ and $S_m$ indicate the smallest (the one-site scale) and largest (lattice) sizes. At criticality, the cumulative form scales as a power law

$$P_>(S) \sim S^{-\tau+1}. \tag{3.10}$$

In our model, the shapes of the distributions change at increasing values of $\varepsilon$: from single-scaled to scale-free, as shown in figure 5*c* (original model) and figure 5*d* (terraformed system). In both cases, the same pattern is found, but the power-law behaviour is observed at much higher levels of soil degradation. Interestingly, the scaling exponent for both cases is the same: $\tau \approx 1.75$, in agreement with the results reported in Kéfi *et al.* [46]. The consideration of the extra processes tied to terraformation do not alter

a universal behaviour pattern, thus suggesting that, despite the parameter shift in $\varepsilon_c$, the warning signals associated with both scenarios will behave in a similar way.

# 4. Discussion

The impact of anthropogenic-driven processes is rapidly increasing, leading to a degradation of extant ecosystems and, in many cases, pushing them closer to viability thresholds [1,2,72,73]. In the case of semiarid ecosystems, this degradation is produced mostly by global warming and intensive grazing [22]. The increasing pressure could end up in catastrophic transitions that are essentially irreversible. It is expected that the expansion of arid areas will increase in the next decades, with an associated increase in the likelihood of green-desert tipping points. Under this potential scenario, it is important to develop strategies of intervention aimed at avoiding these transitions.

Theoretical results on the nature of these shifts and the parameter values at which they take place suggest that some generic factors (such as noise and dispersal [42,43]) could be crucial. Can realistic interventions help preventing green-desert transitions? As discussed above, several bioremediation strategies to face this problem have been proposed. Some recent proposals suggest how to exploit nonlinear features after tipping points based on periodic replanting [19]. More recently, an engineering approach based on synthetic biology has also been suggested [56–58]. In this article, we investigate this latest approach for arid and semiarid ecosystems employing both mathematical and computational models.

Beyond the standard vegetation-desert model, in our study we pay attention to the possibility that an engineered soil crust (a 'synthetic' soil) might have on the fate of fragile or endangered ecosystems. The original motivation is to describe, using a toy model, the impact of two different but complementary strategies: the so-called C-terraformation using a designed cooperative loop (promoted by designed microbial strains); and the D-terraformation, consisting on engineered strains with increased dispersal capacities. Cooperative loops (included with the C-terraformation at the soil crust level in our modelling approach) have indeed been identified in natural ecosystems. For instance, termites activity may modify the nature of transitions to desert states by increasing water infiltration rates and soil moisture [74]. In this manuscript, we have explored the impact of these two synthetic terraformation strategies on the location and behaviour of tipping points.

We show that both terraformation strategies increase the resistance to soil degradation, pushing the tipping points to higher (or even much higher) critical degradation rates ($\varepsilon_c$). This is consistently predicted by both mean-field and spatial models. However, when space is explicitly included, the parameter space supporting this protection against catastrophic shifts is much larger, with a fluctuation dynamics (and underlying warning signals) that behave similarly between the non-engineered and the engineered systems, i.e. they show the same universality pattern.

Our theoretical predictions are limited by the simplifying assumptions that define our model. In particular, the description of soils ignores their development, spatial organization or diversity, as well as the network structure (e.g. trophic relations) of ecosystems [75]. These webs are known to display their own critical thresholds, and thus it remains open how such diverse communities may impact the results reported here. However, despite all these limitations, it is important to highlight that similar previous models of vegetation dynamics in drylands have been very successful in providing deep insight into their natural counterparts [15]. Some key universal properties might pervade the success of these models. In our context, this is likely to be related to the general patterns found close to phase transitions. Additionally, the terraformation scheme described here is likely to be effective when the cooperative interaction helps create the proper ecosystem engineering effect. By engineering ecosystem engineers, endangered arid and semiarid habitats might get protected (at least temporarily) from sudden collapse.

Finally, we may emphasize that ecosystems microbiome is a complex ecosystem itself within the full ecosystem. The introduction of a synthetic species could induce secondary effects at this level that could scale up towards higher ecological (e.g. trophic) levels. However, the synthetic biology approaches suggested here and in previous research [56–58] may re-use already existing microorganisms, which could be synthetically modified. The impact of a truly new species in the ecosystem may be investigated elsewhere by e.g. invasion (adaptive dynamics) or complex networks theory.

Data accessibility. The programming code for the cellular automaton model is provided as a electronic supplementary material.

Authors' contributions. R.V.S. and B.V. conceived the model. B.V. and J.S. wrote the codes of the mean-field model. B.V. wrote the codes of the cellular automaton. All authors analysed the mathematical and computational models. All authors wrote the article and gave final approval for publication.

Competing interests. Authors have no competing interests.

Funding. This study was supported by an European Research Council Advanced Grant (SYNCOM), by the Botin Foundation (Banco Santander through its Santander Universities Global Division), the PR01018-EC-H2020-FET-Open MADONNA project, by the FIS2015-67616-P grant, and by the Santa Fe Institute. This work has also received the support of Secretaria d'Universitats i Recerca del Departament d'Economia i Coneixement de la Generalitat de Catalunya. J.S. has been funded by a 'Ramón y Cajal' contract RYC-2017-22243, and by the MINECO grant no. MTM2015-71509-C2-1-R and the Spain's 'Agencia Estatal de Investigación' grant no. RTI2018-098322-B-I00, as well as by the CERCA Programme of the Generalitat de Catalunya.

Acknowledgements. The authors specially thank Sergi Valverde for the spatial spheric rendering used for the cellular automaton. We also thank Nuria Conde, Joan V., Fernando T. Maestre and Miguel Berdugo for helpful discussions.

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
