## [Reviewer comments · Royal Society Open Science]

Review History

RSOS-200161.R0 (Original submission)

Review form: Reviewer 1

Is the manuscript scientifically sound in its present form?

No

Are the interpretations and conclusions justified by the results?

Yes

Is the language acceptable?

Yes

Do you have any ethical concerns with this paper?

No

Have you any concerns about statistical analyses in this paper?

No

Recommendation?

Major revision is needed (please make suggestions in comments)

Comments to the Author(s)

Please see my report attached (Appendix A).

Decision letter (RSOS-200161.R0)

19-Mar-2020

Dear Dr Sardanes,

The editors assigned to your paper ("Synthetic soil crusts against green-desert transitions: a spatial model") has now received comments from reviewers. We would like you to revise your paper in accordance with the referee and Editor suggestions which can be found below (not including confidential reports to the Editor). Please note this decision does not guarantee eventual acceptance.

Please submit a copy of your revised paper before 10-Apr-2020. Please note that the revision deadline will expire at 00.00am on this date. If we do not hear from you within this time then it will be assumed that the paper has been withdrawn. In exceptional circumstances, extensions may be possible if agreed with the Editorial Office in advance. We do not allow multiple rounds of revision so we urge you to make every effort to fully address all of the comments at this stage. If deemed necessary by the Editors, your manuscript will be sent back to one or more of the original reviewers for assessment. If the original reviewers are not available we may invite new reviewers.

- Ethics statement

- Data accessibility

It is a condition of publication that all supporting data are made available either as supplementary information or preferably in a suitable permanent repository. The data accessibility section should state where the article's supporting data can be accessed. This section should also include details, where possible of where to access other relevant research materials such as statistical tools, protocols, software etc can be accessed. If the data has been deposited in an external repository this section should list the database, accession number and link to the DOI

for all data from the article that has been made publicly available. Data sets that have been deposited in an external repository and have a DOI should also be appropriately cited in the manuscript and included in the reference list.

If you wish to submit your supporting data or code to Dryad (<http://datadryad.org/>), or modify your current submission to dryad, please use the following link:
<http://datadryad.org/submit?journalID=RSOS&manu=RSOS-200161>

- **Competing interests**

- **Authors' contributions**

- **Acknowledgements**

- **Funding statement**

Kind regards,

Andrew Dunn

on behalf of Prof Pete Smith (Subject Editor)

Associate Editor comments:

The journal has had to approach an unusually large number of reviewers, and we've made this decision on the basis of the one (seemingly thorough) report we have received. Please ensure that you fully engage with the queries and comments made by the reviewer - both in a revised manuscript with tracked changes, and a point-by-point response document. If you do not supply these, the admin team will return the paper to you. Furthermore, you will be required to satisfy the reviewer that you have made sufficient changes to meet their requirements - if you do not do

so, we cannot guarantee the paper will be considered further, as the journal only permits one round of revision under normal circumstances.

Reviewers' Comments to Author:

Reviewer: 1

Comments to the Author(s)

Please see my report attached.

Author's Response to Decision Letter for (RSOS-200161.R0)

See Appendix B.

RSOS-200161.R1 (Revision)

Review form: Reviewer 1

Is the manuscript scientifically sound in its present form?

Yes

Are the interpretations and conclusions justified by the results?

Yes

Is the language acceptable?

Yes

Do you have any ethical concerns with this paper?

No

Have you any concerns about statistical analyses in this paper?

No

Recommendation?

Accept as is

Comments to the Author(s)

I appreciate the effort of the authors to address all my comments.

Decision letter (RSOS-200161.R1)

Dear Dr Sardanes,

It is a pleasure to accept your manuscript entitled "Synthetic soil crusts against green-desert transitions: a spatial model" in its current form for publication in Royal Society Open Science.

The comments of the reviewer(s) who reviewed your manuscript are included at the foot of this letter.

Kind regards,

Anita Kristiansen
Editorial Coordinator

on behalf of Pete Smith (Subject Editor)
openscience@royalsociety.org

Reviewer comments to Author:
Reviewer: 1

Comments to the Author(s)
I appreciate the effort of the authors to address all my comments.

Appendix A

Dear Editor of the Journal of Royal Society Open Science,

Thank you very much for the invitation to review the manuscript *RSOS-200161 -- Synthetic soil crusts against green-desert transitions: a spatial model* by Vidiella et al. Please find my comments below.

In this study, Vidiella and coauthors propose two different ways of manipulating arid ecosystems in order to change their dynamical responses to environmental changes. More specifically, the authors study whether (1) facilitating microbial dispersal or (2) introducing new microbial species that facilitate vegetation growth may change the properties of the arid-to-desert transition and the level of aridity at which it occurs. This is definitely an important question in which mathematical modeling and theoretical predictions may play an important role. However, I have important concerns about the study that I would like the authors to clarify.

Comments about the Introduction.

The study is, in general well-motivated, and the Introduction is easy to read. However, I think that there are some points that need to be clarified.

- I do not agree with the fact that facilitation is mediated by non-trophic interactions. What happens if facilitation is mediated, for instance by increased soil quality due to organic matter deposition or litter?
- Concepts such that *catastrophic transitions* or *critical state* need to be defined the first time they are used.
- In lines 34-36 of the Introduction, the authors claim that “*a common outcome of facilitation is the emergence of spatial patchiness. Such patterns are often remarkably organized in space*”. I think that this sentence is not totally accurate. Regular patchiness (i.e., regular patterns) can be entirely mediated by long-range competition, without invoking facilitative interactions (see Martínez-García 2013¹). I agree that many models suggest that a scale-dependent feedback (i.e., the combination of short-range facilitation and long-range competition) is responsible for patterns, but as far as I can tell, there is no empirical evidence of patterns being actually driven by these two interactions acting simultaneously.
- I find the *C-terraformation* very thought-provoking. However, I think the way it is discussed in the manuscript is too simplified. There are many examples of natural ecosystem engineers that improve the local conditions for vegetation growth and thus impact the nature of transitions in the ecosystem dynamics. For instance, Bonachela et al., 2015², showed that termite mounds may modify the nature of transitions to desert states similarly to *C-terraformation* by increasing water infiltration rates and soil moisture. I think the connection between terraformation strategies and naturally occurring ecosystem engineers should be discussed in the manuscript.
- Finally, I wonder that the manipulation of the ecosystem microbiome in all the ways proposed in this work may alter the many other features of the ecosystem dynamics. I think that possibility should be mentioned too. Given the complexity exhibited by soil, microbial communities, how possible it is to introduce a new species without inducing any secondary effect in the ecosystem?

¹ Martínez-García, R., Calabrese, J. M., Hernández-García, E., & López, C. (2013). Vegetation pattern formation in semiarid systems without facilitative mechanisms. *Geophysical Research Letters*, *40*, 6143–6147.

² Bonachela, J. A., Pringle, R. M., Sheffer, E., Coverdale, T. C., Guyton, J. A., Caylor, K. K., Levin, S. A., & Tarnita, C. E. (2015). Termite mounds can increase the robustness of dryland ecosystems to climatic change. *Science*, *347*(6222), 651–655.

Comments about the Models and Results.

My main concern is about the model details. I understand that the authors are working with an extension of an already published framework to which they add two additional mechanisms. However, I think that level of detail that the authors provide about the model description is insufficient which makes the results impossible to replicate. For instance:

- In section II many variables are introduced without being defined or specifying how they can be calculated. For instance, how do you calculate the local density of vegetation ρ_v ? Why is it constant if there is a birth/death dynamics for the vegetation biomass? What is ρ_s ? (line 126).
- I think there is some issue with the dimensions of the equations for the transitions probabilities too. For instance, if $P(D \rightarrow S)$ is a probability, then r cannot be a rate. I think the authors should talk about transition rates (probability per unit of time). The same applies to $P(S \rightarrow D)$ and ε .
- I am also confused by the equation for $P(V \rightarrow S)$. If it is a process that occurs at constant rate m (again problems with the units), then decay in the number of vegetated patches should be exponential, not linear.
- The fact that the authors limit the $0 < \varepsilon < 1$ for simplicity needs to be justified. Do they think that affects in any qualitative or quantitative way any of their conclusions? I think this might be an important point because the authors draw conclusions like: “*The yellow region indicates that vegetation extinction cannot be achieved because $\varepsilon = 1$* ” (caption Figure 3).
- Also in this section, the authors refer to Fig. 1c, when I believe they want to refer to Fig. 1e.

Without having full details about the model implementation, it is hard to absorb all the results. I therefore think that the study will benefit a lot from a careful rewriting of Section II.

In this same line, the mean field limit of the model is given without providing any detail about the calculations that bring the spatial model to this limit. For instance, why is δ not present in Eqs. (2)-(4)? I assume it is some sort of spatial parameter that disappears in the mean field limit, but that should be mentioned. I suggest that the authors include an Appendix with details on the derivation of the mean field limit of the stochastic spatial model.

Regarding the cellular automata model, I found it weird that a new definition of the transition probabilities is introduced without making any connection to those in Section II. I would be important that the authors establish the link between one and the other and why they present them twice. As it is written now, that is not very clear.

In line 379 (page 7) the authors say that “This effect, taking into account that the lattice has...is probably introduced by space more than by stochasticity”. Could the authors be more precise in that state? Maybe running a quick simulation test would be enough to support that statement and make it less speculative.

Finally, I think that most background about the distribution of cluster size has to be provided. What is the physical meaning of S_c ? References for the use of that specific shape of $P(S)$ should be provided^{3,4}. Very recently, a theoretical study also related vegetation patterns to percolation models⁵.

³ Scanlon, Todd M., et al. "Positive feedbacks promote power-law clustering of Kalahari vegetation." *Nature* 449.7159 (2007): 209-212.

⁴ Staver, A. Carla, et al. "Spatial patterning among savanna trees in high-resolution, spatially extensive data." *Proceedings of the National Academy of Sciences* 116.22 (2019): 10681-10685.

⁵ Martín, Paula Villa, Virginia Domínguez-García, and Miguel A. Muñoz. "Intermittent percolation and the scale-free distribution of vegetation clusters." *arXiv preprint arXiv:1912.08369* (2019).

Comments on the figures.

- The title of Figure 1 is rather vague. What do the authors mean by “complexity of semiarid ecosystems”? How do they define “local states”?
- From Fig. 1B I cannot really see the difference between bare soil and soil crust, which I don't think is something easy to evaluate visually. Hence, I question the usefulness of such panel. I would appreciate if the authors could please clarify this point.
- The loops in Fig. 1e are not very informative, and the caption is too vague. Could the authors make sentences like “*Moreover, the presence of vegetation cover has an impact on these transitions, as indicated by the links ending in black circles*” more precise by specifying, for instance the type of impact that vegetation has?
- I agree with previous Reviewers that the circular shape of the snapshots of the spatial distribution is not very clear (but I understand that is up to authors personal preferences).
- The caption of Figure 3 is confusing. Panel C does not even have a caption and I would suggest that the authors use the wording to refer to Panels B in Fig. 2 and 3. Now they say it is “*Fraction of the states at equilibrium increasing and using a full vegetated system as initial conditions*” in Fig. 2 whereas they call it a bifurcation diagram in Fig. 3. I also think that some subtitle for each of the *a, b, c, d* subpanels would be very informative for the reader.
- The labels of some panels in Fig. 4 are very hard or impossible to read, especially those of the stochastic trajectories followed by vegetation and fertile soil with time. That figure has different fonts for *x* and for *x* and *y* axis too.

Typos and References

The manuscript has some typos that I have been able to identify. I list them below together with some parts of the text in which the authors could make their arguments stronger by providing missing references.

- Page 2, lines 64-67. That wording of that sentence sounds a little weird to me.
- Page 3, line 106. “the most fundamental interactions” sounds very general and vague.
- Caption of Figure 2 (fourth line). “display” -> displayed.
- Page 6, line 302. The word “Fig.” before 2(B) is missing.
- Page 7, line 341. “in order to test its robustness” -> I would suggest using model/results instead of “its”.
- Page 7, line 347. I would suggest using transitions among them instead of “their transitions”
- Page 7, line 356. Could the authors have some reference to support that statement?
- Page 7, line 358. I would say “possibly leading to spatial patterning” since these may not always form.
- Page 8, line 406. “have revealed” -> has revealed.
- Eq. (10) has a typo. *S* is the integrating variable so it should be $P_{>}(S_m)$.
- Page 9, line 494. “if” -> is.
- Page 10, line 557. “These are multispecies communities...”. What does this “these” refer to?

Appendix B

Responses letter for manuscript *RSOS-200161*, with title *Synthetic soil crusts against green-desert transitions: a spatial model*, submitted to the Royal Society Open Science.

We want to thank the Editor for handling our manuscript and to the Referee for the exhaustive report. Find please below our answers and a detailed explanation of the changes and corrections introduced in the revised version. For the sake of clarity our responses are indicated below in blue.

Referee comments:

In this study, Vidiella and coauthors propose two different ways of manipulating arid ecosystems in order to change their dynamical responses to environmental changes. More specifically, the authors study whether (1) facilitating microbial dispersal or (2) introducing new microbial species that facilitate vegetation growth may change the properties of the arid-to-desert transition and the level of aridity at which it occurs. This is definitely an important question in which mathematical modeling and theoretical predictions may play an important role. However, I have important concerns about the study that I would like the authors to clarify.

Comments about the Introduction.

The study is, in general well-motivated, and the Introduction is easy to read. However, I think that there are some points that need to be clarified.

- I do not agree with the fact that facilitation is mediated by non-trophic interactions. What happens if facilitation is mediated, for instance by increased soil quality due to organic matter deposition or litter?

We completely agree. The previous explanation wanted to emphasise that biotic agents (organisms) change the environment (abiotic conditions such as soil organic carbon) facilitating the growth of another organism. To highlight this point we have included in the revision the reference Rodriguez-Iturbe *et al.* (2019), which provides a good example where the vegetation creates islands of fertile soil and good moisture conditions for the growing of new plants.

- Concepts such that catastrophic transitions or critical state need to be defined the first time they are used.

Corrected. We have done so in the first paragraph of the introduction.

- In lines 34-36 of the Introduction, the authors claim that “a common outcome of facilitation is the emergence of spatial patchiness. Such patterns are often remarkably organized in space”. I think that this sentence is not totally accurate. Regular patchiness (i.e., regular patterns) can be entirely mediated by long-range competition, without invoking facilitative interactions (see Martinez-Garcia 2013). I agree that many models suggest that a scale-dependent feedback (i.e., the combination of short-range facilitation and long-range competition) is responsible for patterns, but as far as I can tell, there is no empirical evidence of patterns being actually driven by these two interactions acting simultaneously.

We entirely agree with the Referee. Our sentence is not totally accurate. Indeed, we think that it is very difficult to think in ecosystems in which individuals only cooperate (by means of facilitation), since they probably will also undergo some degree of competition and vice versa. We have rewritten this text (now starting in line 24), including other references to help understanding what we wanted to say, and talking specifically in different research concerning only competition (as suggested by the Referee) or both competition-facilitation processes.

- I find the C-terraformation very thought-provoking. However, I think the way it is discussed in the manuscript is too simplified. There many examples of natural ecosystem engineers that improve the local conditions for vegetation growth and thus impact the nature of transitions in the ecosystem dynamics. For instance, Bonachela et al., 2015, showed that termite mounds may modify the nature of transitions to desert states similarly to C-terraformation by increasing water infiltration rates and soil moisture. I think the connection between terraformation strategies and naturally occurring ecosystem engineers should be discussed in the manuscript.

We find this point really interesting and thanks for this reference. We have commented this point in terms of cooperative loops in the third paragraph of the Discussion.

- Finally, I wonder that the manipulation of the ecosystem microbiome in all the ways proposed in this work may alter the many other features of the ecosystem dynamics. I think that possibility should be mentioned too. Given the complexity exhibited by soil, microbial communities, how possible it is to introduce a new species without inducing any secondary effect in the ecosystem?

This is a very good point that we have included in a last paragraph in the Discussion section.

Comments about the Models and Results.

My main concern is about the model details. I understand that the authors are working with an extension of an already published framework to which they add two additional mechanisms. However, I think that level of detail that the authors provide about the model description is insufficient which makes the results impossible to replicate. For instance:

- In section II many variables are introduced without being defined or specifying how they can be calculated. For instance, how do you calculate the local density of vegetation ρ_v ? Why is it constant if there is a birth/death dynamics for the vegetation biomass? What is ρ_s ? (line 126).

We are sorry about the confusion. It is true that ρ_v (and ρ_s) are variables. We have corrected these mistakes and clarified throughout the text the variables and other mathematical terms, especially in Section II. We have also added a piece of text at the beginning of Section II explaining why we are introducing the models and approaches (transitions, mean field model, CA model) in the order found in the article. We think that this is the more natural to introduce the models and the results in Sections II and III. First we present the transitions from a probabilistic manner. Then we study the differential equations model, finally translating the transition rules into an spatial embedding for the CA model. We think that now this issue is more understandable.

- I think there is some issue with the dimensions of the equations for the transitions probabilities too. For instance, if $P(D \rightarrow S)$ is a probability, then r cannot be a rate. I think the authors should talk about transition rates (probability per unit of time). The same applies to $P(S \rightarrow D)$ and e . I am also confused by the equation for $P(V \rightarrow S)$. If it is a process that occurs at constant rate m (again problems with the units), then decay in the number of vegetated patches should be exponential, not linear.

We understand the confusion, but these are probabilities of transition from a state to another one for a given patch, not the global dynamics of the system. $P(V \rightarrow S)$ means the probability that

a given patch of vegetation to transition towards a fertile state. Yes, the decay is exponential. We have revised section II to correct and clarify all these points.

- The fact that the authors limit the $0 < \epsilon < 1$ for simplicity needs to be justified. Do they think that affects in any qualitative or quantitative way any of their conclusions? I think this might be an important point because the authors draw conclusions like: "The yellow region indicates that vegetation extinction cannot be achieved because $\epsilon = 1$ " (caption Figure 3).

Epsilon is a probability and thus must be ranged between 0 and 1. The revision of Section II and section III now clarifies this issue.

- Also in this section, the authors refer to Fig. 1c, when I believe they want to refer to Fig. 1e.

Corrected

Without having full details about the model implementation, it is hard to absorb all the results. I therefore think that the study will benefit a lot from a careful rewriting of Section II.

It is true that the order of appearance of the models (stochastic, ODEs, CA) is a little bit, but we think it is the more natural way of introducing it. Typically, ODEs models go first and then one usually explores the stochastic dynamics, having in background the dynamics predicted by the same system without noise (given by the ODEs)

In this same line, the mean field limit of the model is given without providing any detail about the calculations that bring the spatial model to this limit. For instance, why is d not present in Eqs. (2)-(4)? I assume it is some sort of spatial parameter that disappears in the mean field limit, but that should be mentioned.

We agree that in the text it could be confusing. But as it is written in the article, "The parameter δ balances the influence of the local and global vegetation to produce the germination of new plants in a given site.". In order to emphasise this, we have changed the first sentence where δ appears, to avoid confusions.

I suggest that the authors include an Appendix with details on the derivation of the mean field limit of the stochastic spatial model.

The derivation can be found in the reference:

Vidiella, Blai, Josep Sardanyés, and Ricard Solé. Exploiting delayed transitions to sustain semiarid ecosystems after catastrophic shifts. *J. Royal Society Interface* 15.143 (2018): 20180083.

The new functions (the ones related with the soil crust terraformation) only modify the rates at which the transitions between states occur. We think that with all the changes introduced is not necessary to include the derivation in an Appendix.

Regarding the cellular automata model, I found it weird that a new definition of the transition probabilities is introduced without making any connection to those in Section II. I would be important that the authors establish the link between one and the other and why they present them twice. As it is written now, that is not very clear.

The transition probabilities displayed in Fig. IIIB are the same than the ones presented in Section II, but considering the neighbours (influence of local spatial correlations). The new terms introduced due to this approach are immediately explained.

In line 379 (page 7) the authors say that “This effect, taking into account that the lattice has...is probably introduced by space more than by stochasticity”. Could the authors be more precise in that state? Maybe running a quick simulation test would be enough to support that statement and make it less speculative.

This statement, which we agree that is speculative, is saying that, due to the large system's size (40.000 sites is a very large number), the changes in the shape of Fig. 3A may be due to the local nature of spatial correlations more than due to stochastic effects. Under this system size we know from all the simulations we have carried out that stochastic fluctuations are very weak. Note that the time series in Fig. 4 are shown in very narrow y-axis ranges. The visualization of the trajectories within the y-axis range of [0,1] show very flat curves. Below we plot different time series for vegetation (upper) and fertile soil (lower) for three different lattice sizes: 20 x 20 (green), 50 x 50 (blue), and 200 x 200 (black, the size we used in our simulations to avoid large stochastic effects). Notice the scales in the y-axes are still quite narrow.

We think that it is not worth to put this information in the manuscript (main or supplementary) because the effect of system's size in the fluctuations is very well-known.

Finally, I think that most background about the distribution of cluster size has to be provided. What is the physical meaning of S_c ? References for the use of that specific shape of $P(S)$ should be provided^{3,4}. Very recently, a theoretical study also related vegetation patterns to percolation models⁵.

We thank the referee for providing these references, which will be added to the revised version. We have added the following:

"The cut-off defines the crossover from between a regime of critical clusters (power-law distributed) to that of non-critical clusters. As we get closer to percolation points, the system also displays divergent correlation lengths and $S_c \rightarrow \infty$ thus

rapidly increasing the cutoff value and moving closer to pure power laws. Since the lattice is finite, we will always observe a rapid decay at the end of the distribution. These distributions can be systematically estimated from vegetation data (*references suggested below have been added here*)"

³ Scanlon, Todd M., et al. "Positive feedbacks promote power-law clustering of Kalahari vegetation." *Nature* 449.7159 (2007): 209- 212.

⁴ Staver, A. Carla, et al. "Spatial patterning among savanna trees in high-resolution, spatially extensive data." *Proceedings of the National Academy of Sciences* 116.22 (2019): 10681-10685.

⁵ Martín, Paula Villa, Virginia Domínguez-García, and Miguel A. Muñoz. "Intermittent percolation and the scale-free distribution of vegetation clusters." *arXiv preprint arXiv:1912.08369* (2019).

We have also realised that we used letter *S* to denote the cluster sizes, which could be confused with state *S* of the CA model. We have changed the one for the cluster sizes with a calligraphic *S*.

Comments on the figures.

- The title of Figure 1 is rather vague. What do the authors mean by “complexity of semiarid ecosystems”? How do they define “local states”?

We have removed the title of Fig. 1 by another one more precise. As mentioned above, we have entirely rewritten the caption of this figure.

- From Fig. 1B I cannot really see the difference between bare soil and soil crust, which I don't think is something easy to evaluate visually. Hence, I question the usefulness of such panel. I would appreciate if the authors could please clarify this point.

See the answer in the next comment below.

- The loops in Fig. 1e are not very informative, and the caption is too vague. Could the authors make sentences like “Moreover, the presence of vegetation cover has an impact on these transitions, as indicated by the links ending in black circles” more precise by specifying, for instance the type of impact that vegetation has?

We have rearranged Fig. 1, including more information on the nature of the interactions (whether they are positive or negative (inhibition)). Also, we have clarified how this interactions may occur. We have also removed the small rectangles indicating the different states (vegetation, bare soil, and fertile soil). The Referee is right, they were very difficult to differentiate. Now in panel (b) we have included a detail of the oil crust, typical from semiarid ecosystems. Also, we have first included the panel with the schematic interactions in (c), just before the image of the cellular automaton (now shown in panel (d)).

- I agree with previous Reviewers that the circular shape of the snapshots of the spatial distribution is not very clear (but I understand that is up to authors personal preferences).

In our opinion, the spherical automata representation is visually appealing. However, as we mentioned in the first revision, we added in Fig. 4 representative examples of the spatial patterns obtained with our simulations.

- The caption of Figure 3 is confusing. Panel C does not even have a caption and I would suggest that the authors use the wording to refer to Panels B in Fig. 2 and 3. Now they say it is “Fraction of the states at equilibrium increasing ” and using a full vegetated system as initial conditions” in Fig. 2 whereas they call it a bifurcation diagram in Fig. 3. I also think that some subtitle for each of the a, b, c, d subpanels would be very informative for the reader.

We have rewritten the caption of this figure.

- The labels of some panels in Fig. 4 are very hard or impossible to read, especially those of the stochastic trajectories followed by vegetation and fertile soil with time. That figure has different fonts for x and for x and y axis too.

We have changed the colour for the Fertile soil and increased the size of the fonts to make them more readable. We now provide a vertical figure instead of a landscape-like one to ease visualisation.

Typos and References

The manuscript has some typos that I have been able to identify. I list them below together with some parts of the text in which the authors could make their arguments stronger by providing missing references.

- Page 2, lines 64-67. That wording of that sentence sounds a little weird to me.

We entirely agree. We have rewritten this sentence (now in lines 84-87). Focusing on what we are exploring in our contribution.

- Page 3, line 106. "the most fundamental interactions" sounds very general and vague.

We have arranged this sentence to be more precise (now lines 125-1

- Caption of Figure 2 (fourth line). "display" -> displayed.

Corrected

- Page 6, line 302. The word "Fig." before 2(B) is missing.

Corrected

- Page 7, line 341. "in order to test its robustness" -> I would suggest using model/results instead of "its".

Corrected

- Page 7, line 347. I would suggest using transitions among them instead of "their transitions"

Corrected

- Page 7, line 356. Could the authors have some reference to support that statement?

Sure, we have added some key references on this point.

- Page 7, line 358. I would say "possibly leading to spatial patterning" since these may not always form.

This sentence as was written was ambiguous, since no references were added. We have added the references in which spatial patterning in semiarid ecosystems is shown and studied, thus our statement does not need the word "possibly". We want to thank the Referee for raising this point.

- Page 8, line 406. "have revealed" -> has revealed. - Eq. (10) has a typo. S is the integrating variable so it should be $P > (S_m)$.

Corrected.

The cumulative distribution is denoted as $P>$, which is a function of the cluster sizes, S .

- Page 9, line 494. "if" -> is.

Corrected

- Page 10, line 557. "These are multispecies communities...". What does this "these" refer to?

We have clarified this point, which was unclear and misleading.

Authors comment: All the references have been formatted with the RSOS style.